# Effects of Imported Semen Based on Different Selection Indices on Some Production and Reproduction Traits in Iranian Holstein Cattle

**DOI:** 10.3390/ani12213054

**Published:** 2022-11-07

**Authors:** Masume Nazari, Peyman Mahmoudi, Amir Rashidi, Mohammad Razmkabir

**Affiliations:** Department of Animal Science, University of Kurdistan, Sanandaj 6617715175, Kurdistan, Iran

**Keywords:** breeding value, bull semen, Holstein cattle, Lifetime Net Income index

## Abstract

**Simple Summary:**

We evaluated the impact of imported semen of Holstein bulls from different countries on their daughters’ economic traits using the Lifetime Net Income (LNI) index in different climates of Iran. The first lactation records were collected from 10 big dairy farms in various regions of Iran. Studied traits included milk, fat and protein yields, calving age and calving interval. There were significant differences between sire groups for studied traits in each climate (*p* < 0.01). In semi-cold, moderate and warm climates, daughters of French sires had the highest least-square means of LNI index. In cold climate, daughters of German sires had the highest least-square means.

**Abstract:**

The aim of the present study was to evaluate the effects of imported semen of Holstein bulls from different countries on the economic traits of their daughters using the Lifetime Net Income (LNI) index in various climates of Iran. The data included the first lactation records of 274,057 Holstein cows collected during 1993 to 2017 by the Animal Breeding Center of Iran from 10 large dairy farms located in various provinces of Iran. The investigated traits included milk, fat and protein yields, calving age and calving interval. Breeding values of progenies were predicted by the Best Linear Unbiased Prediction (BLUP) method under the multi-trait animal model using DMU software. The genetic-economic merit of the progenies was estimated by the LNI index. There were significant differences between the estimated breeding values (EBVs) of sire groups (based on bull semen origin) for milk, fat and protein yields, calving age and calving interval in each climate (*p* < 0.01). The obtained results showed that the highest least-square means of LNI index in semi-cold, moderate and warm climates belonged to the daughters of French sires; however, daughters of German sires were estimated to have the highest least-square means in the cold climate.

## 1. Introduction

For many years, dairy cattle breeding programs have mainly been directed toward the improvement of production traits. This has been true for most countries, except for Scandinavian countries, whose selection indices have also included health and reproduction, and North American countries, whose selection indices have included conformation traits [1]. On the other hand, the improvement of productive traits has caused adverse effects on cattle reproduction, resulting from a negative correlation between productive and reproductive traits [2,3]. In general, increasing economic efficiency is the objective of any breeding program; thus, in addition to productive traits, it is necessary to consider reproductive traits for developing an efficient breeding program.

Selection index is a theory for maximizing genetic progress to achieve maximized profitability [4]. Since 1942, much progress has been made in the field of dairy cattle breeding. Selection indices are important tools in conventional dairy cattle breeding because they incorporate information about several traits into a single value that can be used to rank animals and make the best selection decisions [5]. Most countries have their selection index in accordance with their production and economic systems [1]. A major difference between the selection indices in various countries is the relative emphasis on productive traits [1,6,7,8].

In the Holstein cattle population, genetic improvement through the selection of males compared to females is of crucial importance [9]. Importing semen from approved countries is one of the most effective and common tools for genetic improvement of economic traits of Holstein cattle in developing countries such as Iran. According to Animal Breeding Center of Iran data, semen importation to Iran from countries such as the United States, Canada and several European countries, particularly Germany and the Netherlands, began in 1964 and still continues. About 60 to 80% of imported semen are from American and Canadian sires [10]. The application of imported semen may have various outcomes under environmental conditions of countries where these semen are used. For these reason, genetic evaluation of bulls providing semen in exporting countries may not be appropriate to predict the performance of their progenies in importing countries [11]. The difference between actual and expected performance of the progenies resulting from imported semen is one of the most important concerns of breeding programs due to the specific environmental characteristics of the importing country [12]. Taking into consideration that Iran has a variety of weather and climatic zones and considering that there are a lot of countries around the world with similar conditions to Iran, the main purpose of the present study was to evaluate the effects of Holstein bulls’ semen imported from different approved countries on the economic traits of their daughters using the Lifetime Net Income (LNI) [13] index in various climates of Iran.

## 2. Materials and Methods

### 2.1. Preparation of Data

Data used in the present study included the first lactation records of 274,057 Holstein cows from 1993 to 2017 provided by the Animal Breeding Center of Iran (ABCI). The origin of sires was from the United States, Canada, France, the Netherlands, Italy and Germany. The investigated traits included milk, fat and protein yields, calving age and calving interval. Cows with less than 1000 kg and more than 15,000 kg of total milk production, herds with fewer than 30 records and first calving age less than 18 months and more than 42 months were excluded from data.

Records were classified under cold, semi-cold, moderate and warm climates based on the climatic data for stations where data collected [14]. Accordingly, Iran’s climates were divided into four groups based on the average annual temperatures of 11.5, 13.5, 16.1 and more than 19.5 °C (Table 1). Furthermore, sire origins of investigated cows were classified into six groups based on the country, including Canada, the United States, the Netherlands, France, Germany and Italy.

### 2.2. Estimation of Breeding Values

Breeding values of progenies were estimated by the Best Linear Unbiased Prediction (BLUP) methodology under multi-trait animal models using DMU software [15]. The multi-trait animal model was as follows (Equation (1)):(1)$$y_{i}=X_{i}b_{i}+Z_{i}a_{i}+e_{i}$$
where $y_{i}$ is the vector of records for the *i*th trait; $b_{i}$ is the vector of fixed effects for the *i*th trait (in Iran: herd, year-season and climate; in each climate: herd and year-season); $a_{i}$ is the vector of random effects of animal for the *i*th trait; $e_{i}$ is the vector of residual effects for the *i*th trait; and $X_{i}$ and $Z_{i}$ are incidence matrices assigning records to the fixed and random effects of animal, respectively.

### 2.3. Data Description for Estimation of Genetic-Economic Merits

Data from 10 dairy farms situated in different regions of Iran, where the majority of industrial dairy farms are situated, were utilized to calculate economic weights of traits. The production and reproduction records of cattle in all farms were tracked by ABCI. Performance of the cows was retrieved from a large collection of data given by ABCI that was collected by farmers through a survey, or it was calculated using cost and revenue modeling. Data sources used for deriving economic input parameters were based on the marketing circumstances in 2017 and 2018.

### 2.4. Profit Function Model

Under Iranian dairy farm systems, the main dairy herd population comprises four major types of animals including calves, heifers, cows (both milking and dry), and fattened calves. Because it is related to feedlot and cattle production, the last group was excluded from the profit function. Profit, determined as total income minus total costs, was selected as the economic assessment criterion for the stated production system. Profit was expressed in terms of per cow calving per year as the reference unit. Iran’s currency is the Rial (Rl). Henceforth, the costs are stated in the US dollar, assuming an exchange rate of USD 1 = IRR 42,000. The following is the total profit function (Equation (2)):(2)$$P=\sum_{i=1}^{3}p_{i}=\sum_{i=1}^{3}\left(R_{i}-C_{i}\right)$$
where P is total herd profit in dollars per cow per year; R, C and $p_{i}$ are revenues, costs and profits, respectively, from the given animal group per animal per year; i = 1 stands for female and male calves from birth to 3 months of age or death; i = 2 represents heifers from 3 months of age to either age at first calving, selling, culling or death; and i = 3 stands for cows. Costs were divided into feed and non-feed costs. Non-feed costs included labor, veterinary, breeding, housing, fuel and insurance costs. Feed costs were based on energy and protein requirements.

### 2.5. Gene Expression

The impacts of genes expression may be observed as genetic development in diverse characteristics across time in various animal populations. Due to the potential cost of delayed gains, it is usually desired for genetic supremacy to be shown as soon as possible. This research updated the gene flow approach initially described by [16] for beef cattle to quantify genes expression for many classes of traits in Iranian Holstein dairy cattle. For sires of self-replacing females, the gene expression was calculated according to [13].

### 2.6. Calculation of Economic Weights

According to [17], the economic values (EVs) of a trait is the change in profit per unit change of that trait, assuming no change in other traits. Economic weights (EWs) were calculated by multiplying the number of expressed genes by EV. Since a zero-discount rate was utilized, the EW did not consider the different time when the traits were expressed throughout the life of animal [18].

#### 2.6.1. Milk Production Traits

The first derivative of the profit function (Equation (1)) was selected considering milk, fat and protein yields for which the EV was required. Since there are no supply quotas for dairy farms in Iran, this is essentially equal to collecting the difference between the profits and costs of producing 1 kg of each product. Expenses associated with milk production were calculated by taking into account feed and labor, breeding, veterinary, fuel, housing and insurance costs. Only expenses related to metabolizable protein and net energy of lactation requirements were taken into account for protein and fat yields [18]. The estimated energy requirements to yield 1 kg of milk protein and butterfat were 5.47 and 9.29 Mcal, respectively. Furthermore, 1.47 kg of metabolizable protein in feed and 1 kg of net protein in milk were needed for producing 1 kg of net protein in milk [19].

#### 2.6.2. Age at First Calving

The EV for age at first calving was estimated according to [13]. In addition, the requirements for transitioning a heifer weighing 625 kg with conceptus and entering first lactation was calculated according to [19], allowing more 20% to cover feeding costs. Rearing expenses were adjusted for sex ratio, stillbirth, calving rate, heifer mortality and pre- and post-weaning survival rates, representing the EV per cow per year. In addition, the EW of age at first calving was calculated according to [13].

#### 2.6.3. Calving Interval

The EV of calving interval was calculated as the first partial derivative of the total profit function (Equation (1)), with respect to calving interval. Since cows express the trait just once throughout calving intervals and as a result have a relative gene expression coefficient of unity, the EW of calving interval is equivalent to the EV.

#### 2.6.4. Evaluation of Genetic-Economic Merit

In the present study, the EW was estimated to be USD −0.051 per kilogram of milk yield; USD 4.881 per kilogram of fat yield; USD 3.985 per kilogram of protein yield; USD −0.916 per day of age at first calving; and USD −1.695 per calving interval.

The genetic-economic merit of 274,057 cows in Iran and its various climates was estimated using the LNI index as follows (Equation (3)):Lifetime Net Income (LNI) $ = ($−0.051 × breeding value of milk yield) + ($4.881 × breeding value of fat yield) + ($3.985 × breeding value of protein yield) − ($0.916 × age at first calving) − ($1.695 × the first calving interval)(3)

### 2.7. Analysis of Estimated Breeding Values and LNI Index

The models used to analyze estimated breeding values (EBVs) and LNI index in Iran (Equation (4)) and each climate (Equation (5)) were as follows:(4)$$y_{ijklm}=\mu +H_{i}+YS_{j}+R_{k}+I_{l}+{\left(R\times I\right)}_{kl}+e_{ijklm}$$
(5)$$y_{ijkm}=\mu +H_{i}+YS_{j}+C_{k}+e_{ijkm}$$
where *y_ijklm_* and *y_ijkm_* are observation, *μ* is the overall mean, *H_i_* is the fixed effect of *i*th herd, *YS_j_* is the fixed effect of *j*th year-calving season, *R_k_* is the fixed effect of *k*th climate, *I_l_* is the fixed effect of lth selection index, (*R × I*)*_kl_* is the interaction between *k*th climate and lth selection index, *C_k_* is the fixed effect of *k*th sperm exporting country and *e_ijklm_* and *e_ijkm_* are random residual effects.

The comparison of EBVs and LNI index of the sire groups was carried out by Tukey test using SAS 9.2 software [20]. Finally, the best sire groups were determined in terms of genetic-economic merit in Iran and its various climates.

## 3. Results and Discussion

Descriptive statistics of investigated traits is presented in Table 2. The number of records for milk, fat and protein production was 269,786, 216,914 and 270,566 records, respectively. In total, 270,410 and 186,898 records were available for age at first calving and calving interval, respectively. Cows in moderate climate had the highest average of milk, fat and protein production, while the lowest mean for age at first calving and calving interval belonged to cows recorded from cold climate.

The comparisons of EBVs for production traits between cows from different sire groups in Iran and cold, semi-cold, moderate and warm climates are shown in Table 3. The effect of the bull semen source was significant on the performance of their daughters regarding milk, fat and protein production in Iran and its climates (*p* < 0.01). Daughters of French sires had the highest EBVs for production traits (milk, fat and protein yields) in Iran and semi-cold, moderate and warm climates. In the cold climate, the highest EBVs for milk and fat yields belonged to daughters of American sires, and for protein yield, the highest EBVs were estimated for daughters of German sires.

The Canadian selection indices have always had a high relative emphasis on the production traits [1,8]; however, in the present study, the Canadian sires had low EBVs for these traits in Iran and the considered climates. The breeding values of foreign bulls are estimated based on their progenies’ performance under different environmental and climatic conditions of the same country. Therefore, those conditions may not be matched completely with the conditions in the other countries and different climates destinations. For this reason, there is often a difference between the expected performance and the actual performance of progenies under new conditions [21]. Therefore, in Iran and its climates, it seems that possible genotype by environmental interactions had a high influence on genetic expression of production traits for Canadian sires. In a study, authors of [22] reported the EBV means of milk yield for sires from the United States, Canada, the Netherlands and New Zealand as 231, 120, −66 and 15 kg, respectively. Dastanian and his colleagues [23] reported the EBV means of milk yield for American, Canadian and European sires as 732, 500 and 511 kg, respectively. Powell and Wiggans [24] reported that EBV mean of milk yield for daughters of the United States sires (born in 1985) was 380 kg, which was higher than that of daughters of Canadian sires, 336 kg.

Comparison of EBVs for reproduction traits between daughters from different sire groups in Iran and cold, semi-cold, moderate and warm climates is shown in Table 4. The effect of the bull semen source was significant on the calving age and calving interval of their daughters in Iran and its climates (*p* < 0.01). In Iran and semi-cold, moderate and warm climates, the daughters of French sires had the highest EBVs for the first calving age, and the daughters of German sires had the highest EBVs for the calving interval. Under cold climate, the daughters of German sires and French sires had the highest EBVs for the first calving age and calving interval, respectively. Since 2000, France included reproductive traits in selection indices of dairy cows in addition to the production traits. Thus, from 2001 to 2019, the relative emphasis on reproductive traits increased from 12.5% to 22%. Moreover, from 1997, selection indices in Germany have directly been focused on reproductive traits. However, the extent of this emphasis has always been relatively low so that the relative emphasis on these traits is only 10% today [1,8]. Applying a multi-trait selection index in which the genetic progress can be simultaneously obtained for several traits is reported as the most effective way for reducing the decline in the efficiency of reproductive traits [25].

Comparison of least-square means of LNI index for the studied production and reproduction traits between cows from different sire groups in Iran and cold, semi-cold, moderate and warm climates is shown in Table 5. In Iran and semi-cold, moderate and warm climates, the highest least-square means of LNI index belonged to daughters of French sires, while it was highest in cold climates when the semen of German sires were used. These findings revealed that French sires had the highest genetic-economic merit for studied traits in Iran and semi-cold, moderate and warm climates. In Iran and its climates, daughters of French sires had the highest EBVs for protein yield, but EBVs for the first calving age were lowest. In addition, daughters of French sires had the highest EBVs for fat yield. Taking into consideration the positive EWs for fat and protein yields in the LNI index, as well as the negative EW for the first calving age in this index, the estimated genetic-economic merit for French sire was the highest. Moreover, in cold climate, the daughters of German sires had the highest genetic-economic merit for the studied traits, while these daughters had the lowest genetic-economic merit for these traits in the semi-cold climate. In cold climate, these cows had the highest EBVs for protein and fat yields, in addition to the highest EBVs for first calving age. On the other hand, all of these factors had a direct and positive impact on the genetic-economic merit of cows under the mentioned climate. Semen exportation is used in other countries in different circumstances (e.g., payment system, feed costs and feed supply); thus, there is likely a reduction in the expected economic responses. For instance, Holmann and colleagues [26] evaluated the profit of investing in semen from American Holstein bulls for using in Colombia, Mexico and Venezuela and suggested that importations should be used strategically in environments with excellent feed and management in order to obtain a positive response on the selection of imported genetic material.

As a result, it seems necessary to develop a specific genetic-economic selection index for each climate of Iran, and breeding and economic objectives of different regional climates should be considered.

## 4. Conclusions

There were significant differences between the EBVs for milk, fat and protein yields, the first calving age and calving interval of sire groups (semen origins) in Iran and its various climates. In Iran, the highest and lowest EBVs belonged to French and German sires, respectively. In addition, the daughters of French sires in moderate climate had the highest EBVs regarding milk production, while the best performance of fat and protein production belonged to daughters of German sires in moderate and daughters of French sires in semi-cold climates, respectively. The lowest age at first calving was observed in calves of French sires in moderate climates, and lowest calving interval belonged to daughters of German sires in warm climates. Furthermore, in Iran and semi-cold, moderate and warm climates, the highest least-square means of the LNI index of progenies (genetic-economic merit) were estimated for French sires and in cold climate for German sires. As a result, it seems necessary to develop a specific genetic-economic selection index in each climate of Iran, and we recommended semen be imported based on breeding and economic objectives of different climates.

## Figures and Tables

**Table 1 animals-12-03054-t001:** Classification of provinces based on climatic conditions as the definition of environment.

Climate	Province	Area (%)	Temperature (°C)		
			Average	Minimum	Maximum
Cold	Ardabil, Chahar Mahaal and Bakhtiari, East Azerbaijan, Hamadan, Kurdistan, Markazi, Qazvin, West Azerbaijan, Zanjan	13.2	11.0	−3.0	23.6
Semi-cold	Alborz, Mazandaran, North Khorasan, Tehran	12.9	13.5	0.0	25.7
Moderate	Gilan, Isfahan, Kermanshah, Khorasan Razavi, Lorestan	21.2	16.1	2.9	28.2
Warm	Bushehr, Fars, Golestan, Hormozgan, Ilam, Kerman, Khuzestan, Kohgiluyeh and Boyer-Ahmad, Qom, Semnan, Sistan and Baluchestan, South Khorasan, Yazd	52.7	23.7	11.4	34.7

**Table 2 animals-12-03054-t002:** Descriptive statistics of investigated traits.

Trait	Climate	*n*	Mean	SD	Minimum	Maximum	CV
Milk (kg)	Iran	269,786	9627.47	1620.94	2519.06	14,999.59	20.99
	Cold	64,775	9597.92	1631.16	2519.06	14,999.59	21.16
	Semi-cold	117,303	9654.46	1601.25	2570.33	14,998.65	19.69
	Moderate	75,127	9718.34	1745.62	2571.55	14,998.35	22.27
	Warm	12,581	8985.37	1764.72	2648.38	14,953	22.77
Fat (kg)	Iran	216,914	258.37	48.37	50.52	478.37	18.72
	Cold	46,016	263.55	50.27	50.52	468.24	19.08
	Semi-cold	108,664	252.88	45.76	55.64	475.69	18.09
	Moderate	55,028	266.03	49.25	51.85	471.51	18.51
	Warm	7138	249.88	55.25	81.52	478.37	22.11
Protein (kg)	Iran	270,566	256.92	63.53	51.35	479.86	24.73
	Cold	64,875	250.02	63.06	52.52	479.39	25.22
	Semi-cold	117,633	262.79	62.42	58.11	479.86	23.75
	Moderate	75,449	256.59	65.01	51.35	479.64	25.34
	Warm	12,609	239.82	60.26	77.52	473.71	25.13
Age at first calving (month)	Iran	270,410	25.28	2.96	20.00	41.97	11.71
	Cold	64,789	25.14	2.90	20.00	41.93	11.53
	Semi-cold	117,443	25.16	2.85	20.03	41.97	11.33
	Moderate	75,605	25.37	2.94	20.00	41.93	11.60
	Warm	12,573	26.56	3.91	20.03	41.97	14.74
Calving interval (day)	Iran	186,898	394	70.51	300	600	17.87
	Cold	44,446	395	70.51	300	600	17.88
	Semi-cold	80,271	395	71.10	300	600	18.01
	Moderate	54,187	389	68.44	300	600	17.60
	Warm	7994	401	71.10	300	600	17.73

SD: Standard deviation. CV: Coefficient of variance.

**Table 3 animals-12-03054-t003:** Comparison of estimated breeding values (EBVs) for production traits between daughters from different sire groups in Iran and its various climates (kg).

Trait/Sire Groups	Iran	Climate				*p*-Value
		Cold	Semi-Cold	Moderate	Warm	
Milk						<0.001
the United States	649.45 ^b^	633.72 ^a^	686.50 ^b^	608.70 ^b^	607.10 ^b^	
Canada	467.16 ^d^	503.35 ^b^	482.17 ^d^	421.78 ^c^	475.35 ^c^	
France	769.80 ^a^	379.42 ^c^	815.40 ^a^	837.51 ^a^	761.13 ^a^	
Netherlands	571.65 ^c^	615.00 ^a^	457.62 ^d^	637.48 ^b^	611.59 ^b^	
Italy	574.79 ^c^	588.17 ^ab^	566.90 ^c^	623.94 ^b^	440.36 ^c^	
Germany	365.68 ^e^	527.23 ^b^	224.23 ^e^	573.87 ^b^	254.67 ^d^	
Fat						<0.001
the United States	11.56 ^b^	11.26 ^a^	11.86 ^b^	11.21 ^c^	11.54 ^b^	
Canada	8.70 ^e^	8.30 ^c^	8.14 ^d^	9.87 ^d^	9.70 ^c^	
France	12.76 ^a^	7.71 ^c^	12.91 ^a^	13.25 ^b^	14.20 ^a^	
Netherlands	11.35 ^b^	10.08 ^b^	9.08 ^cd^	14.62 ^a^	13.35 ^a^	
Italy	10.30 ^c^	9.73 ^b^	9.52 ^c^	12.51 ^b^	10.35 ^c^	
Germany	9.39 ^d^	10.38 ^ab^	5.67 ^e^	14.63 ^a^	13.89 ^a^	
Protein						<0.001
the United States	10.20 ^c^	9.55 ^c^	10.85 ^b^	9.90 ^d^	8.99 ^c^	
Canada	8.69 ^d^	9.03 ^d^	8.67 ^c^	8.47 ^e^	8.66 ^c^	
France	16.11 ^a^	9.01 ^d^	17.99 ^a^	15.99 ^a^	16.61 ^a^	
Netherlands	11.81 ^b^	11.99 ^b^	10.13 ^b^	13.65 ^b^	11.34 ^b^	
Italy	8.99 ^d^	8.82 ^d^	7.68 ^c^	11.58 ^c^	9.40 ^c^	
Germany	7.27 ^e^	15.97 ^a^	4.17 ^d^	9.03 ^de^	12.11 ^b^	

EBVs with a different letter within columns are significantly different (*p* < 0.01).

**Table 4 animals-12-03054-t004:** Comparison of estimated breeding values (EBVs) for reproduction traits between daughters from different sire groups in Iran and its various climates (day).

Trait/Sire Groups	Iran	Climate				*p*-Value
		Cold	Semi-Cold	Moderate	Warm	
Calving age						<0.001
the United States	−20.35 ^b^	−19.48 ^b^	−22.24 ^b^	−18.54 ^d^	−17.24 ^b^	
Canada	−16.09 ^c^	−20.69 ^b^	−16.04 ^c^	−13.27 ^e^	−14.41 ^b^	
France	−46.35 ^a^	−4.23 ^d^	−51.26 ^a^	−54.15 ^a^	−37.93 ^a^	
Netherlands	−10.81 ^d^	−10.96 ^c^	−7.57 ^e^	−13.96 ^e^	−12.69 ^bc^	
Italy	−16.84 ^c^	−20.64 ^b^	−14.49 ^c^	−21.07 ^c^	−7.50 ^c^	
Germany	−15.91 ^c^	−31.08 ^a^	−9.80 ^d^	−25.85 ^b^	17.38 ^d^	
Calving interval						<0.001
the United States	3.45 ^d^	3.44 ^c^	3.58 ^d^	3.24 ^b^	3.53 ^c^	
Canada	3.15 ^c^	2.90 ^b^	3.38 ^d^	3.05 ^b^	2.93 ^b^	
France	2.71 ^b^	1.64 ^a^	2.38 ^c^	3.43 ^b^	2.70 ^b^	
Netherlands	3.51 ^d^	3.67 ^c^	3.32 ^d^	3.56 ^b^	3.33 ^bc^	
Italy	2.62 ^b^	2.12 ^b^	1.95 ^b^	4.63 ^c^	3.03 ^b^	
Germany	1.17 ^a^	2.78 ^b^	0.90 ^a^	1.30 ^a^	0.81 ^a^	

EBVs with a different letter within columns are significantly different (*p* < 0.01).

**Table 5 animals-12-03054-t005:** Comparison of least-square means of Lifetime Net Income (LNI) index for production and reproduction traits between daughters from different sire groups in Iran and its various climates (USD).

Sire Groups	Iran	Climate			
		Cold	Semi-Cold	Moderate	Warm
the United States	76.62 ^b^	72.59 ^b^	80.29 ^b^	74.50 ^d^	70.89 ^c^
Canada	62.58 ^c^	64.77 ^c^	58.56 ^c^	67.33 ^d^	65.76 ^c^
France	124.94 ^a^	55.22 ^d^	135.88 ^a^	129.31 ^a^	126.71 ^a^
Netherlands	77.16 ^b^	69.32 ^bc^	57.69 ^c^	99.88 ^b^	85.03 ^b^
Italy	67.66 ^c^	67.75 ^bc^	58.02 ^c^	86.72 ^c^	67.17 ^c^
Germany	68.68 ^c^	111.08 ^a^	40.27 ^d^	99.50 ^b^	85.74 ^b^

Least-squares means of LNI with different superscripts within columns are significantly different (*p* < 0.01).

## Data Availability

The data presented in this study are available on request from the corresponding author. The data are not publicly available due to privacy and ethical restrictions.

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
