# Peer review of "Effects of Imported Semen Based on Different Selection Indices on Some Production and Reproduction Traits in Iranian Holstein Cattle"

_animals, 2022, doi:10.3390/ani12213054_

Round 1

Reviewer 1 Report

The manuscript "Evaluation of imported semen effects based on different selection indices on economic traits of Iranian Holstein cattle in various climates", by Nazari et al. deals with a topic of great importance in the animal sciences.

In fact, genetic selection programs are generally implemented on a national basis, but the spread of instrumental insemination allows the use of selected males on a worldwide basis.

The manuscript, therefore, attempts to investigate the well-known, but difficult to estimate, GxE interaction.

Even if the paper is quite well written, the scientific approach could be improved, in particular:

1) by better illustrating some lesser-known passages (e.g. paragraphs 2.3 to 2.6) and streamlining some more well-known and less directly related to the presented work (e.g. to illustrate the estimates of the breeding values, it would be sufficient to write the model in scalar form and eliminate the rest);

2) the number of analyzed data (and their distribution based on the variability factors taken into consideration) is never reported, neither in materials and methods nor in the tables. This does not allow for an appropriate results evaluation;

3) furthermore, it is not clear to me why the Tukey test was used to analyze continuous variables and, since the opposite has not been stated, I imagine normally distributed.

It is not clear what model of comparison was used, from what is written it would seem a one-way analysis of variance ?

If that's the case, it's not the best way to analyze the presented data. You should use a model that considers the two sources of variability and their interaction.

In any case, the model must be formalized exactly in the text. 

Author Response

We highly appreciate your comments towards improving our manuscript.

Point 1: by better illustrating some lesser-known passages (e.g., paragraphs 2.3 to 2.6) and streamlining some more well-known and less directly related to the presented work (e.g., to illustrate the estimates of the breeding values, it would be sufficient to write the model in scalar form and eliminate the rest);

Response 1: Thank you for your comment. The suggestions by the reviewer have been applied. Please see section 2.2 in the revised manuscript.

Point 2: the number of analyzed data (and their distribution based on the variability factors taken into consideration) is never reported, neither in materials and methods nor in the tables. This does not allow for an appropriate results evaluation;

Response 2: Thank you for your suggestion. The summary statistics of data is added to the manuscript. Please see Table 1 in the revised manuscript.

Point 3: furthermore, it is not clear to me why the Tukey test was used to analyze continuous variables and, since the opposite has not been stated, I imagine normally distributed. It is not clear what model of comparison was used, from what is written it would seem a one-way analysis of variance? If that's the case, it's not the best way to analyze the presented data. You should use a model that considers the two sources of variability and their interaction. In any case, the model must be formalized exactly in the text.

Response 3: The models used for analyzing data have been added to the manuscript. Please see section 2.7 in the revised manuscript.

Reviewer 2 Report

The authors investigate the effects of semen of Holstein bulls from various countries on the important traits (milk, fat and protein yields, calving age and calving interval) for their daughters in Iran based on Lifetime Net Income (LNI) index in different climates. The data mainly indicated that different semen sources under the background of climates and geographical locations have different performances on the economic traits of female Iranian Holstein cattle. These are very important for improving the production benefits and utilization factor.  Overall, the manuscript is fine.  However, I have major concerns as follows:

1. Gene microarrays or high throughput sequencing strateges in screening out economic-related positions of animal genomic variations are now broadly applied in animal breeding. If the background of climates and geographical locations the were used to combine with sequencing data, will it help to find out the main character in controlling phenotype from genetic perspective?

2. The written contents in this draft should be carefully checked, some duplicates were found. e.g. Line 207-211, these two sentences seems duplicated and the values were not consistence. 

3. Tables should be presented more clearly. e.g. Table 2.3.4, P values in each clolumn should be added. Or else, too many letters may confused the readers. Are all P<0.01? 

4. Data Availability Statement are not declared.

Author Response

We highly appreciate your comments toward improving our manuscript.

Point 1: Gene microarrays or high throughput sequencing strategies in screening out economic-related positions of animal genomic variations are now broadly applied in animal breeding. If the background of climates and geographical locations the were used to combine with sequencing data, will it help to find out the main character in controlling phenotype from genetic perspective?

Response 1: Using genomic data, relationships between individuals are more accurately estimated given that the markers can account for Mendelian inconsistencies and for the contemporary and historical pedigree, if the number of markers is enough to path the identical-by-descent (IBD) status across the genome. Genomic data reveals more genetic connectedness than pedigree data, because animals likely share at least some alleles, and this has been shown to increase the accuracy of genetic evaluation. On the other hand, some breeding programs, especially in developing countries, are ineffective as a consequence of the non-adaptability of the introduced breeds to the challenging environments. For instance, Ferreira et al. (2017) and Rosé et al. (2017) showed that differences between temperate and tropical climates can cause significant genotype by environment interaction (G×E), which affects productivity. Combining sequencing data and geolocation information can aid in predicting in which regions a breed will most probably exhibit an environmental mismatch. It can also reveal potential areas for successful introduction of animals, contributing to a successful breeding program. In summary, this allows making predictions of breed-specific suitability taking into account location information. Being able to explain the role of G×E can be a useful application that will further help in providing support when designing breeding programs, or introduction programs for local animal production, by understanding the environmental variables that can have an impact on breed productivity between geographical locations.

Point 2: The written contents in this draft should be carefully checked, some duplicates were found. e.g., Line 207-211, these two sentences seem duplicated and the values were not consistence.

Response 2: Thank you for this comment. The statement has been revised.

Point 3: Tables should be presented more clearly. e.g., Table 2.3.4, P values in each column should be added. Or else, too many letters may confuse the readers. Are all P<0.01?

Response 3: Yes, all comparisons were significant at the level of 0.01. The p-Values were added to Tables as suggested by the reviewer.

Point 4: Data Availability Statement are not declared.

Response 4: Thank you. Data Availability Statement was added.

References:
Ferreira JL, Lopes FB, Garcia JAS, Silva MPB, Nepomuceno LL, Marques EG and Silva MCD. 2017. Climate spatialization and genotype-environment interaction effects on weaning weights of Nellore cattle in extensive systems in tropical regions of Brazil. Ciência Animal Brasileira 18, 18.
Rosé R, Gilbert H, Loyau T, Giorgi M, Billon Y, Riquet J, Renaudeau D and Gourdine JL. 2017. Interactions between sire family and production environment (temperate vs. tropical) on performance and thermoregulation responses in growing pigs. Journal of Animal Science 95, 4738–4751.

Reviewer 3 Report

The conclusion is short compared to the large number of results presented in the research. That should be corrected.

Author Response

Thank you for your comment. The conclusion section has been improved.

Round 2

Reviewer 1 Report

no other comments